# CRIDECO Anticholinergic Load Scale: An Updated Anticholinergic Burden Scale. Comparison with the ACB Scale in Spanish Individuals with Subjective Memory Complaints

**DOI:** 10.3390/jpm12020207

**Published:** 2022-02-03

**Authors:** Hernán Ramos, Lucrecia Moreno, Jordi Pérez-Tur, Consuelo Cháfer-Pericás, Gemma García-Lluch, Juan Pardo

**Affiliations:** 1Cátedra DeCo MICOF-CEU UCH, Universidad Cardenal Herrera-CEU, CEU Universities, 46115 Valencia, Spain; ramgarher@alumnos.uchceu.es (H.R.); lmoreno@uchceu.es (L.M.); jpereztur@ibv.csic.es (J.P.-T.); m.consuelo.chafer@uv.es (C.C.-P.); gemma.garcia2@alumnos.uchceu.es (G.G.-L.); 2Department of Pharmacy, Universidad Cardenal Herrera-CEU, CEU Universities, 46115 Valencia, Spain; 3Institut de Biomedicina de València-CSIC, CIBERNED, Unitat Mixta de Neurologia i Genetica, IIS La Fe, 46010 Valencia, Spain; 4Alzheimer Disease Research Group, Health Research Institute La Fe, 46026 Valencia, Spain; 5Embedded Systems and Artificial Intelligence Group, Universidad Cardenal Herrera-CEU, CEU Universities, 46115 Valencia, Spain

**Keywords:** cognitive impairment, anticholinergic burden scale, anticholinergic drug, subjective memory complaint

## Abstract

The increase in life expectancy has also been accompanied by an increase in the use of medication to treat chronic diseases. Polypharmacy is associated with medication-related problems such as the increase in the anticholinergic burden. Older people are more susceptible to anticholinergic effects on the central nervous system and this, in turn, may be related to cognitive impairment. In this paper, we develop an updated anticholinergic burden scale, the CRIDECO Anticholinergic Load Scale (CALS) via a systematic review of the literature and compare it with the currently most used Anticholinergic Burden Scale (ACB). Our new scale includes 217 different drugs with anticholinergic properties, 129 more than the ACB. Given the effect that anticholinergic medications have on cognitive performance, we then used both scales to investigate the relationship between anticholinergic burden and cognitive impairment in adult Spanish subjects with subjective memory complaint. In our population, we observed an association between cognitive impairment and the anticholinergic burden when measured by the new CALS, but not when the ACB was applied. The use of a more comprehensive and upgraded scale will allow better discrimination of the risk associated with the use of anticholinergic medications on cognitive impairment. CALS can help raise awareness among clinicians of the problems associated with the use of medications, or combinations of them, with large anticholinergic effect, and promote a better personalized pharmacological approach for each patient.

## 1. Introduction

Life expectancy is one of the most important measures for assessing the overall health of a population and data indicate a very positive trend in recent years. Spain is the fourth country with the highest life expectancy after Japan, Switzerland and the Czech Republic [1]. This increase in life expectancy has also been accompanied by an increase in the appearance of age-related chronic diseases and the concomitant use of medications to treat them. This situation is reflected in the increase in polypharmacy in recent years, considering the term polypharmacy as the consumption of five or more medicines [2]. The prevalence of polypharmacy in our country, according to the latest studies, is 27.3% [2].

Polypharmacy is associated with increased drug–drug interactions, increased risk of adverse drug reactions [3], decreased adherence to treatment [4], increased risk of frailty [5], increased risk of hip fractures [6] and increased risk of falls and hospitalizations [7]. In this context, it is important to note that the use of drugs with anticholinergic properties has increased in recent years [8].

Anticholinergic medications are drugs that block the neurotransmitter acetylcholine from binding to muscarinic receptors [9]. The cumulative effect of taking drugs with anticholinergic actions is known as the anticholinergic burden [10]. On the other hand, the effects of anticholinergic drugs are divided into peripheral (dry mouth, blurred vision, constipation, tachycardia and urinary retention) and central (writing confusion, dizziness and even cognitive impairment) [9]. Moreover, older people are more susceptible to anticholinergic effects on the central nervous system due to a higher permeability of the blood–brain barrier [11]. The estimated prevalence of the use of these drugs ranges from 12.5% [12] to 49.5% [13] depending on the population and the scale used.

Within the muscarinic receptors, there are different subtypes, M1 to M5. M1 receptors are the most common receptors in the central nervous system and play important roles in executive abilities and episodic memory in the hippocampus and prefrontal cortex [14]. Furthermore, antagonism of M2 (memory processing) and M4 receptors (acetylcholine level regulation) can lead to cognitive disorders and cell death [15]. This leads to a direct link between cognitive impairment (CI) and the use of anticholinergic drugs. Numerous studies associate the use of anticholinergic drugs with CI [16,17,18,19], and several longitudinal studies even consider the use of these drugs as a risk factor for dementia [20,21,22,23]. In addition, the use of these drugs has been linked to higher mortality [24,25] and increased risk of hospitalizations [26].

However, the existence of many scales measuring the anticholinergic burden means that the conclusions drawn can be variable. Moreover, many of them are more than a decade old, such as the Anticholinergic Drug Scale (ADS) or the Anticholinergic Cognitive Burden (ACB) scale, and continue to obtain the highest percentages in quality today [27]. In addition, the ACB scale is currently the most widely used and includes 88 drugs with an anticholinergic effect [27]. Continued drug approvals by regulatory agencies make it necessary to update these scales, as many now commonly used anticholinergic drugs are not included.

Alzheimer’s disease (AD) has now been shown to be a biological continuum between the early asymptomatic (preclinical AD), mildly symptomatic (subjective cognitive decline) or moderately symptomatic (mild cognitive impairment) stages and the most severe phase (dementia) [28]. The term subjective cognitive decline (SCD) was conceived to describe cognitively intact individuals who are concerned about declining cognitive function [29]. In fact, SCD has been classified as the stage prior to mild cognitive impairment (MCI) in individuals with positive AD biomarkers according to the National Institute of Ageing research framework [28]. Over the last decade, knowledge of SCD has increased enormously, positioning it as the earliest symptomatic manifestation of AD [30]. Subjective memory complaint (SMC) is one of the criteria necessary to include patients in SCD, along with a normal score on cognitive tests used to classify MCI or prodromal AD [31]. Furthermore, SMC has been linked to a higher incidence of dementia, doubling the likelihood of developing the pathology [32].

This implies that studies carried out in the population with SMC will not discriminate well between risk and protective factors for CI compared to those studies carried out in the general population. This is the case of anticholinergic drugs, where we hypothesize that the anticholinergic burden scales currently in use, as they only include a part of the existing anticholinergic drugs, may not differentiate well the risk of CI related to the use of these drugs in a special sample.

The aim of this paper is the development of an updated anticholinergic burden scale through a systematic review of the literature and the inclusion of new drugs. In addition, this new scale will be applied to adult Spanish subjects with SMCs, in order to explore the relationship between anticholinergic treatment and CI.

## 2. Materials and Methods

### 2.1. Systematic Review

A systematic review was conducted in PubMed and Web of Science databases to identify the main anticholinergic scales used following PRISMA guidelines [33]. This review was carried out between 6 and 10 October 2021. The keywords used were: “review” AND “anticholinergic” AND “burden” AND “scale”. The search and review process are presented in Figure 1. Papers that did not include anticholinergic burden scales or were not written in English were excluded. Manuscripts that presented lists of drugs with an assigned score were included. A total of 167 records were first identified, 69 in PubMed and 98 in Web of Science. After removing any duplicates, 110 records were selected. For the title-based screening, those that related anticholinergic use to other pathologies different from CI, or did not mention any tool or scale, were eliminated. Of the remaining 45 records, only those that included the use of an anticholinergic scale in their abstract were selected. Thus, sixteen full-text articles were assessed for eligibility.

From them, three manuscripts whose scoring system was not comparable to other scales were excluded [34,35,36]. Two identified manuscripts used the same scales as those already selected in their original articles and were therefore also excluded [37,38]. Finally, three manuscripts that did not include the list of drugs used [27,39,40] and one manuscript not based on expert opinion were also excluded [41].

The final list of manuscripts included seven scales: ADS, Anticholinergic Risk Scale (ARS), ACB, Duran Scale (DS), Salahudeen Scale (SS), German Anticholinergic Burden Scale (GABS) and Korean Anticholinergic Burden Scale (KABS) [42,43,44,45,46,47,48].

### 2.2. Development of a New Anticholinergic Burden Scale

All the tools chosen rated from 1 to 3 the anticholinergic score, except the review by Duran et al., which scores the drugs as high potency (3) or low potency (1 or 2) [45]. In this case, we scored from 1 or 2 when in Durán’s review the different authors gave both scores, and if the score was homogeneous, we scored directly with the value shown in the corresponding column. The same was true for Salahudeen’s review, offering several scores for the same drug [46]. Furthermore, in Durán’s review, drugs with different scores were classified as discrepant (“Disc”).

In the CRIDECO Anticholinergic Load Scale (CALS), we classified the anticholinergic potency of the drugs from 1 (lowest) to 3 (highest). When in the previous scales the score for a particular drug was homogeneous (all of them scored the same drug with the same score) we used such a score. When there were discrepancies between scales, we chose the mean value rounding to the nearest whole number.

The Medicines Database of the Spanish General Council of Pharmaceutical Associations was used to review the medicines commercialized in Spain [49]. Topical, otic, ophthalmic and nasal forms were excluded due to lower systemic absorption [47]. In the event of doubt, these drugs were studied individually, and their final score was determined by the expert committee. This committee was composed of a professor of pharmacology, a clinical pharmacist and a community pharmacist. The pharmacists based their final decisions on their clinical experience and a review of the current literature.

### 2.3. Subject Recruitment and Data Collection

Recruitment was conducted in 19 community pharmacies in the Valencian region (Spain) over 18 months (September 2018 to March 2020). Patient recruitment ended with the onset of the SARS-CoV-2 pandemic in Spain. The protocol used was the same as in previous studies [50,51].

All community pharmacists were trained by the project’s neurologists. During the dispensing routine, the pharmacist identified, either by express reference from the patient, by a close relative or by direct observation by the pharmacist, signs of SMC or possible cognitive impairment, depressive feelings, increased sleepiness, impaired speech or object recognition, difficulty in performing complex activities such as using public transport, problems with money management or medical treatment. Subjects who met the inclusion criteria were informed of the study in the community pharmacies. Inclusion criteria were age 50 years or older, SMCs and willingness to participate. Conversely, the exclusion criteria were age less than 50 years, no SMCs, diagnosis of any dementia, severe sensory deficits (blindness, deafness) and physical disability that could interfere with the performance of the tests.

This study was reviewed and approved by the Institutional Review Board (IRB) of Universidad CEU Cardenal Herrera (CEI18/027, date of approval: 2 February 2018) and by the IRB of the Arnau de Vilanova Hospital (MOR-ROY-2018–013, date of approval: 18 July 2018). All subjects gave written informed consent in accordance with the Declaration of Helsinki.

The variables used were collected through a personalized interview at the community pharmacy, which lasted approximately 40 min. Moreover, the questionnaire used included additional lifestyle variables and dietary habits. The Anatomical Therapeutic Chemical (ATC) code of the World Health Organization (WHO) Collaborating Centre for Drug Statistic Methodology was used to classify the drugs [52]. When a combination was identified, it was considered as such, instead of considering active ingredients separately. Nevertheless, an exception was made with 12 ATC codes (N06CA01, N02BE51, N02BA51, H02BX93, R05DA20, M01AE51, R01BA52, R05CA10, C07CB03, R05FA02, N04BA03 and C03EB01). The above-mentioned ATC codes represent several combinations with diverse active ingredients, each of which has a different anticholinergic burden. In this case, patients taking these codes were identified and the combination ATC codes were separated into individual ATC codes.

### 2.4. Cognitive Impairment Assessment

Patients were assessed using three validated tests: Memory Impairment Screen (MIS), Short Portable Mental Questionnaire (SPMSQ) and Semantic Verbal Fluency Test (SVF). The tests used were chosen on the recommendation of the Valencian Society of Neurology. The idea of using these three tests was to detect as many true positives as possible and thus increase the accuracy of the overall process. Subjects with positive results in at least one of these tests were classified as possible CI patients. With this protocol, the CI confirmation rate in neurology was 90% [50].

#### 2.4.1. Memory Impairment Screen

The MIS is a brief test of memory disorders using free and selectively facilitated recall of four words, giving each word a value of 2 points when recalled without a cue and a value of 1 when a cue is needed (0−8) [53]. It uses controlled learning and selectively facilitated recall techniques. Controlled learning is based on the subject identifying the word to be remembered according to a semantic cue (category of each word). The same semantic cue is used for facilitated recall. The most effective cut-off score is ≤4 points, where the sensitivity for general dementia in the Spanish population was 74% and the specificity 96% [54]. For AD, it obtained sensitivities of 86% and 96%, respectively [54].

#### 2.4.2. Short Portable Mental State Questionnaire (Spanish Version)

The SPMSQ test assesses different aspects of intellectual functioning, including short-term memory, long-term memory, information about current events, orientation and the ability to perform serial mathematical work [55]. It is a short test that is easy to perform and score, with a total of 10 items and a cut-off point of 3 or more errors, does not require specific equipment and applies to illiterate people. The Spanish version of the test showed high sensitivity and specificity for CI, 85.7% and 79.3%, respectively [56].

#### 2.4.3. Semantic Verbal Fluency

The SVF test measures the number of items of a category (in this case animals) that a subject can recall in one minute. This test is very specific for temporal lesions, and in fact has been used preferentially in patients with AD, where there is a progressive alteration of semantic memory, attributed to alterations in the frontal and temporal lobes [57]. Not only is it valid for the detection of patients with AD dementia, but it can also discriminate with high sensitivity between patients with normal cognition, MCI and vascular dementia [58]. The recommended cut-off point is less than 10 points, at which the sensitivity for the detection of general dementia for this test in Spain was 90% and the specificity 94% in a population with a poor educational level [59].

### 2.5. Statistical Treatment

A machine learning technique protocol was used in the community pharmacy to rapidly select candidates for further screening via a question-based CI test [60]. A new artificial intelligence tool, developed by the research group, and based on an ensemble of different machine learning supervised techniques (such as bagging and boosting decision trees) was used to calculate the MCI probability result on performing the tests [60]. The idea was to maximize the selection process by attending to those factors that entail a high probability of positive results in the screening tests. For this reason, the subjective memory complaint was used as a criterion for inclusion in the study. Thus, different statistical and artificial intelligence data analysis techniques were used to examine the collected data.

After completion of the follow-up of the subjects, all the information was stored in a database designed specifically for this study. Subsequently, the data were checked by reviewing the subjects’ medical records during the overall process and a subsequent data cleansing process was conducted to check the completeness and correctness of the dataset. The statistical analysis was carried out using advanced statistical treatment RStudio© (version 4.1.1 (10 August 2021) Integrated Development for R. RStudio, PBC, Boston, MA, USA) [61].

Our initial hypothesis was that the new scoring system, the CALS, would discriminate better than the ACB scale CI for individuals. Thus, the sample size to conduct the study was calculated using G*Power statistical software (version 3.1.9.6, Düsseldorf, Germany) [62]. A two-sample t-test was carried out to check differences between two independent means (two groups). The parameters were set as a two-sided test with medium effect size, power of 0.95 and a significance level of 0.05. This concluded a minimum of 210 subjects and our final total sample was 512 subjects.

Moreover, Chi-squared tests were also conducted to discover if CI is dependent on a relevant (higher than 2) anticholinergic load with the new CALS, and in that sense, a significance level was established at 0.05.

## 3. Results

### 3.1. CRIDECO Anticholinergic Load Scale

Through our systematic review of the literature, we included seven scales: ADS, ARS, ACB, Duran Scale (DS), Salahudeen Scale (SS), GABS and KABS (Figure 2) [42,43,44,45,46,47,48]. Figure 2 shows the selected scales, the number of drugs included in each one, the first author, year of publication and country, together with our new scale. The final CALS included a total of 217 different drugs with anticholinergic properties.

After our systematic review, we selected 125 low potency (Score 1) anticholinergic drugs (Table 1), 28 drugs with medium potency (Table 2) and 62 drugs with high anticholinergic potency (Table 3). Of these, colchicine was ruled out, as one author rated this drug with a score of 1 or 3 [46], another scale rated it as discrepant (Disc) and two scales explicitly rated it with a null anticholinergic score [42,48].

Table 2 shows the 28 drugs with proposed medium potency (Score 2) selected for the CALS. In this case, two drugs were evaluated by the experts: disopyramide and prochlorperazine. Prochlorperazine was maintained with a score of 2 and disopyramide was changed to a score of 1 (Table 4).

Finally, the review selected 62 drugs with high anticholinergic potency (Score 3; Table 3). There were discrepancies in the score of six drugs rated as level 3: cyproheptadine, fluphenazine, maprotiline, olanzapine, paroxetine and perphenazine. All those assessed were changed to level 2 by the team’s pharmacologists, except cyproheptadine, which remained at level 3.

**Table 4 jpm-12-00207-t004:** CRIDECO Anticholinergic Load Scale.

Low Potency (Score 1)				Medium Potency (Score 2)	High Potency (Score 3)	
Aclidinium ^inh^	Cyclosporine	*Iloperidone*	Phenobarbital	Amantadine	*Acepromazine*	*Hyoscyamine*
Alimemazine *	Desloratadine	Ipratropium ^inh^	Piperacillin	Baclofen	Amitriptyline	Imipramine
Alprazolam	Desvelanfaxine	Isosorbide _mononitrate_	Pramipexole	Carbamazepine	*Amoxapine*	Levomepromazine *
*Alverine*	Dexamethasone	Isosorbide _dinitrate_	Prednisolone	Cloperastine	Atropine	Meclozine *
Amisulpride	Dextromethorphan	Ketorolac	Prednisone	*Cimetidine*	*Belladonna*	*Mequitazine*
Ampicillin	Diazepam	Ketotifen	*Pridinol*	Cyclobenzaprine	*Benzatropine**	Nortriptyline
Aripiprazole	*Digitoxin*	Levocetirizine	Pseudoephedrine	*Dosulepin*	Biperiden	*Opipramol*
Asenapine	Digoxin	Levodopa-carbidopa	*Quinidine*	*Fluphenazine*	Brompheniramine	*Orphenadrine*
Atenolol	Diltiazem	Lithium	Risperidone	Loxapine	*Carbinoxamine*	Otilonium bromide
Azathioprine	Dipyridamole	Loperamide	Rotigotine ^patch^	Maprotiline	*Carisoprodol*	Oxybutynin
Benazepril	Disopyramide	Loratadine	Selegiline	Meperidine *	Chlorphenamine *	*Pheniramine*
Betaxolol	Domperidone	Lorazepam	Sertraline	Methadone	Chlorpromazine	Procyclidine
Bisacodyl	Entacapone	*Lumiracoxib*	Sumatriptan	*Molindone*	*Chlorprothixene*	Promethazine
Bromocriptine	Escitalopram	Mebeverine	Tapentadol	*Nefopam*	*Cimetropium bromide*	*Propantheline*
*Bromperidol*	*Estazolam*	Metformin	*Temazepam*	Olanzapine	*Clemastine*	Propiverine
Bupropion	Famotidine	Methocarbamol	Theophylline	Oxcarbazepine	Clomipramine	*Protriptyline*
Captopril	Fentanyl	Methotrexate	*Tiotixene*	Paroxetine	Clozapine	Pyrilamine *
*Cefamandole*	Fexofenadine	Methylprednisolone	Tiotropium ^inh^	Perphenazine	Cyproheptadine	Scopolamine *
Cefoxitin	*Flunitrazepam*	Metoclopramide	Trandolapril	Pimozide	*Darifenacin*	Solifenacin
Celecoxib	Flupentixol	Metoprolol	Trazodone	*Prochlorperazine*	*Desipramine*	*Thioridazine*
*Cephalothin*	Fluoxetine	Midazolam	Triamcinolone	*Promazine*	Dexbrompheniramine	*Tiemonium iodide*
Cetirizine	Flurazepam	Mirtazapine	Triamterene	*Propoxyphene*	Dexchlorpheniramine	*Timepidium bromide*
Cinnarizine	Fluvoxamine	Morphine	Trimebutine	Quetiapine	Dicyclomine *	Tizanidine
Chlordiazepoxide	Furosemide	Naratriptan	Triazolam	Ranitidine	*Difemerine*	Tolterodine
Chlortalidone	Gentamicin	*Nefazodone*	Umeclidinium ^inh^	Tramadol	Diphenhydramine *	*Trifluoperazine*
Citalopram	Glycopyrronium ^inh^	Nifedipine	Valproic acid	Triprolidine	Doxepin	Trihexyphenidyl
Clindamycin	Guaifenesin	*Nizatidine*	Vancomycin	*Zotepine*	Doxylamine	Trimipramine
Clonazepam	Haloperidol	*Oxazepam*	Venlafaxine	Zuclopenthixol	*Emepronium*	*Tropatepine*
Clorazepate	Hydralazine	Oxycodone	Warfarin		Fesoterodine	Trospium
Codeine	*Hydrocodone*	Paliperidone	Ziprasidone		Flavoxate	*Valethamate*
*Cortisone*	Hydrocortisone	*Pancuronium*	Zolmitriptan		*Homatropine*	
*Cycloserine*	Hydromorphone	*Phenelzine*			Hydroxyzine	

Drugs in italics are not currently commercialized/authorized in Spain as of November 2021. * Alimemazine/ Trimeprazine; * Benzatropine/ Benztropine; * Dicyclomine/ dicycloverine; * Diphenhydramine/ dimenhydrinate; * Chlorphenamine/ chlorpheniramine; * Levomepromazine/ Methotrimeprazine; * Meclozine/ meclizine; * Meperidine/ Pethidine; * Pyrilamine/ Mepyramine; * Scopolamine/ Hyoscine. ^inh^ = inhalative.

Table 4 shows the final scale, where medicines not commercialized or not authorized in Spain in November 2021 are shown in italics. The inhaled drug umeclidinium was added to this scale because the rest of the inhaled anticholinergic drugs were included (aclidinium, glycopyrronium, ipratropium and tiotropium) and it is also approved in Spain. The same applies to the drugs tapentadol and hydromorphone because of their pharmacological similarity to other opioids, which are included in Spanish anticholinergic drug bulletins. As in previous studies, the total anticholinergic burden (TAB) was obtained by adding the score of each drug. A score ≥ 3 is considered as a clinically relevant anticholinergic burden [42,43,44,47,48].

### 3.2. Comparison with the ACB and CALS Scale in Individuals with Subjective Memory Complaints

In this study, the participants (*n* = 512) were people aged between 50 and 96 with SMC. They were classified according to tests scores as suggestive of CI (*n* = 164, 32.03%) or cognitively normal (*n* = 348, 67.97%). Table 5 summarizes the demographic and clinical variables of the participants between both groups. Specifically, the TAB score was obtained for each patient on the new scale, as well as in the ACB scale, to compare its association with CI.

The mean TAB obtained with CALS for the non-cognitively impaired group was 1.62 (±1.67), while for the cognitively impaired group it was 2.14 (±1.88), showing statistically significant differences in the total anticholinergic burden between both groups (*p*-value = 0.0026) (Figure 3A). On the other hand, with the ACB scale no significant differences were observed (*p*-value = 0.1439), the mean obtained for the group without CI was 0.87 (±1.37) and 1.08 (±1.53) for the group with CI (Figure 3B).

Next, we analyzed the relationship of the TAB values considered clinically relevant TAB (≥3) and the presence of CI in our population using the two scales (Figure 4). For clinically relevant TAB values, the CALS identified 148 individuals with a relevant anticholinergic score (≥3) and 367 individuals with a lower anticholinergic score (<3). Among patients with CI, 37% had a relevant TAB score, compared to 25% of patients without CI (*p*-value = 0.005). On the other hand, with the ACB scale, 21% of patients with CI had a relevant TAB score, compared to 14% of those without cognitive impairment (*p*-value = 0.0521).

The fifteen most used drugs with anticholinergic effects in the study sample were obtained (Figure 5). These included four benzodiazepines, two long-acting (diazepam and clonazepam) and two intermediate-acting (lorazepam and alprazolam). Five antidepressants, including three selective serotonin reuptake inhibitors (SSRIs) (escitalopram, paroxetine, sertraline), a selective serotonin and norepinephrine reuptake inhibitor (SNRI) (desvenlafaxine) and a serotonin antagonist and reuptake inhibitor (SARI) (trazodone). A minor opioid alone (tramadol) and in combination with other analgesics (paracetamol). An oral antihyperglycemic drug alone (metformin) and in combination with other antidiabetics (sitagliptin and vildagliptin). And finally, a loop diuretic (furosemide).

Furthermore, in our sample, the number of anticholinergic drugs used was significantly associated with CI-compatible scores in any of the three neuropsychological tests (*p*-value = 0.026). Specifically, the mean consumption in the group with CI was 6.52 (±3.31), compared to 5.74 (±3.28) in the group with a normal (or nonimpaired) score in all three tests.

Finally, a total of 38 patients were taking benzodiazepines (N05BA) and opioids (N02A), resulting in a central nervous system (CNS) depressant effect, which can lead to serious side effects such as slow or difficult breathing and death [63].

## 4. Discussion

Anticholinergic drugs form a heterogeneous group comprising active compounds with very diverse indications and applications. Many commonly used medications have anticholinergic effects. It is well known that antimuscarinic drugs can trigger both peripheral and central side effects such as the cognitive alterations that we observed in our population. Several longitudinal studies have linked exposure to various types of anticholinergics to increased risk of dementia [20,21,22,23]. Furthermore, the use of these drugs was associated with increased brain atrophy, dysfunction, and cognitive decline [64]. Similarly, there is a clear link between anticholinergic load and reduced cognitive performance [16,17,18]. Not only do these drugs have an impact on clinical entities such as CI, but they have also been linked to increased risk of falls [7] and mortality [24,25]. Furthermore, concomitant use of several medications with anticholinergic action will further increase the likelihood of adverse events [20]. However, a recent survey of pharmacists revealed that 45% of them did not know that these drugs were a risk factor for dementia [65] and only 44% of healthcare professionals knew that cognition was adversely affected by anticholinergics [66].

Low awareness of the risk of these drugs, together with the diversity of the anticholinergic burden scales, often makes them complex to identify by healthcare professionals. In the present work, we have tried to palliate this deficiency, collating the largest number of drugs with anticholinergic effects identified in the scientific literature. The newly developed scale not only includes drugs already published in previous studies, or authorized after those publications, but has also added three drugs authorized in Spain due to their pharmacological similarity to other drugs with reported anticholinergic effects. The aim is for it to serve as a clinical aid tool for the different health professionals, especially in our country.

Several scales have been designed and published over the years, but certain discrepancies remain among them. One of the main differences among the scales reported so far lies in the definition of whether a drug has an anticholinergic effect and also whether it has a high or low contribution to the global anticholinergic burden. These differences are one of the main sources of confusion when comparing scales [46]. At present, anticholinergic burden classification remains unclear, as the scores of different scales are based on subjective characteristics: bibliography and expert opinion. In addition, the wide variety of associated anticholinergic symptoms makes it difficult to assign a score from 0 to 3. Despite the aforementioned limitations, a recent systematic review suggested that measuring anticholinergic activity based on a literature review was more effective in detecting significant adverse effects than measuring serum anticholinergic activity [67].

Moreover, several have been commercialized or withdrawn from the market by the different regulatory agencies over time. Thus, the updating of the different scales becomes crucial. Therefore, the development of an updated anticholinergic scale compiling as many drugs with anticholinergic activity as possible may be useful.

Our systematic review analysis selected seven anticholinergic scales, whose publication ranged from 2006 to 2019, with the ACB scale being the most widely used to date according to citation analyses [44]. In addition, a recent systematic review comparing 19 scales ranked the ACB and the GABS scale with the highest quality ratings [27]. In fact, these were the reasons why this scale was chosen for comparison with the CALS.

To the best of our knowledge, this scale, with 217 active ingredients, currently includes the largest number of drugs. To minimize the selection bias that could appear by limiting the number of medicines in the scale, we added all the drugs obtained from the literature review, eliminating colchicine only, as two scales attributed no anticholinergic effects to it [42,48] and two other manuscripts assessed it as discrepant [45,46]. With this new, more thorough and inclusive scale, we can calculate the total anticholinergic burden of a particular person and, depending on their general health situation, recommend alternatives to reduce the cholinergic burden.

When comparing two scales (CALS, ACB), we observed that the TAB obtained with the CALS was higher than that calculated with the ACB scale. Moreover, we observed a statistically significant association between either total TAB or clinically relevant TAB and CI, whereas no such associations were seen with the TABs derived from the ACB scale. This finding could be explained by the lower number of drugs with anticholinergic activity in the ACB scale compared to the CALS. We hypothesize that by not including some drugs with an anticholinergic effect recognized by numerous authors, their contribution to this cholinergic risk is lost in some patients.

It should be noted that the study sample is a special population, with patients with subjective memory complaints, so it is to be expected that the prevalence of dementia is higher than in the general population. In fact, the subjective memory complaint, in many patients, may be the first clinical manifestation after some AD histopathological hallmarks (β-amyloid deposition, pathological tau, neurodegeneration), within a biological continuum in Alzheimer’s disease [68]. Considering that anticholinergic drugs have been widely associated with cognitive decline, it is to be expected that a more sensitive scale will allow us to better discriminate the risk associated with this factor in a more specific population.

According to the cholinergic hypothesis, Alzheimer’s disease is due to a decrease in acetylcholine synthesis [69]. Increasing acetylcholine levels by inhibiting acetylcholinesterase is one of the few therapeutic strategies to increase cognition and neural cell function [69]. Given that most of the drugs currently approved for the symptomatic treatment of Alzheimer’s disease are aimed at restoring neuronal cholinergic activity [70], it would be contradictory to prescribe drugs whose pharmacological effect blocks this activity. Furthermore, the use of anticholinergic drugs in patients with dementia has been associated with increased mortality [25]. Therefore, in elderly patients, especially in patients with dementia [71] but also in individuals at risk such as those with subjective memory complaints [32], due to increased susceptibility to anticholinergic effects, the use of these drugs should be carefully evaluated.

Regarding the most prescribed anticholinergic drugs, benzodiazepines, antidepressants, hypoglycaemic drugs and opioids were the most consumed drugs in our population. This coincides with the fact that those same medications are also widely used among elderly patients. The fact that those drugs exhibit low anticholinergic activity (score = 1) individually, except for tramadol and paroxetine (score = 2 for both), reinforces the need for a standardized scale to alert clinicians to avoid co-prescription of these drugs when needed. Furthermore, tramadol is a drug that is not included in the ACB scale, being a widely established drug with an anticholinergic effect [42,45,46,47,48]. In Spain, this drug is extensively used for the treatment of chronic pain, so those patients taking it are only one point away from having a clinically relevant anticholinergic burden.

Niikawa and colleagues reported the existence of an association between polypharmacy and CI [72]. On the other hand, Baek and collaborators and Tapianen and colleagues linked increased consumption of benzodiazepines to cognitive decline [73,74]. This is consistent with our findings in our population, selected by reporting SMC, where an increase in the number and use of anticholinergic drugs prescribed, and the corresponding TAB score, correlated with the presence of CI in our neuropsychological tests.

### Strengths and Limitations

One potential limitation of the CALS is the variability of authorized anticholinergic drugs between countries and the impact this may have on an individual’s final anticholinergic score. To avoid this limitation, we tried to include in this new scale most drugs with reported anticholinergic activity and we did not limit the scale to those compounds authorized in our country but collected information from several international scales, some frequently used in different studies.

Regarding the methodology of medication registration, one strength of the present study is that it was recorded through a personal interview and a review of the centralized electronic prescription registry of individuals. Therefore, drugs prescribed but probably not taken by patients were eliminated when applying the CALS. Because of this, the results obtained do reflect the actual patient’s anticholinergic burden. Another aspect related to medication registration was the ATC code used. We decided to use it to classify active ingredients because it is the international code of drugs classification. Nevertheless, by using it, we identified a limitation when identifying with the same ATC a variable combination of active ingredients. However, this was resolved by taking these combinations as separate active ingredients taken simultaneously.

Another limitation of our study may be found in the specificity of the population under study. Although they all have high sensitivity and specificity for Alzheimer’s disease, bear in mind that these parameters may vary depending on the origin of the CI and the clinical situation of the individual under study. Despite this, it is known from clinical practice that, if a patient has an alteration in any of these tests, it is highly probable that he or she has CI. In addition, these tests are also used in multiple studies as tools to measure CI. In any event, the neuropsychological screening performed does not yield a diagnosis but is an indication of the existence of cognitive problems. In our research project, individuals scoring as likely to have CI are directed to primary or tertiary health providers for full exploration and diagnosis.

## 5. Conclusions

We describe here a new anticholinergic scale that gathers information on 217 active principles, expanding previously existing scales. The new CALS was able to discriminate, more broadly than the ACB scale, the risk association between anticholinergic use and CI. Clinically relevant TAB scores measured with the CALS scale were associated with CI, so the presence of an updated tool in standard clinical practice may be useful, especially when prescribing medication to elderly patients with subjective memory complaints or with a dementia diagnosis. The development of updated scales may help provide us with a better measure of the influence of these drugs on neurodegenerative diseases such as dementia in longitudinal clinical studies. Nevertheless, further research is needed to validate the present scale and future updates will be required. Finally, with the use of this tool, clinicians can be more aware of their prescriptions and make a better personalized pharmacological approach for each patient.

## Figures and Tables

**Figure 1 jpm-12-00207-f001:**
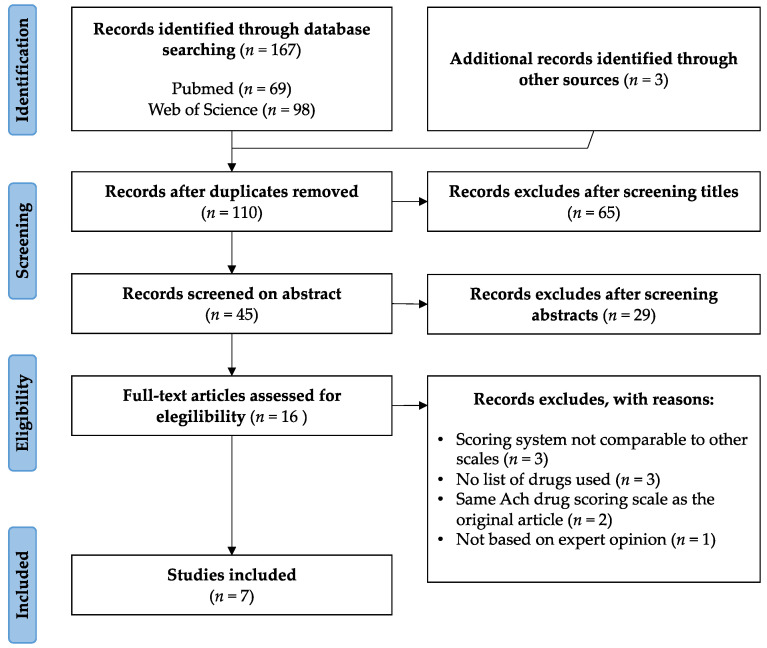
PRISMA flow diagram of study selection.

**Figure 2 jpm-12-00207-f002:**
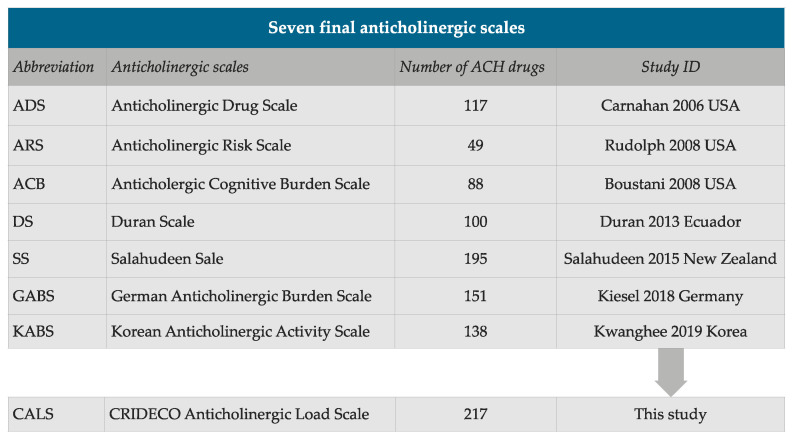
Summary of the number of anticholinergic (ACH) drugs on each scale.

**Figure 3 jpm-12-00207-f003:**
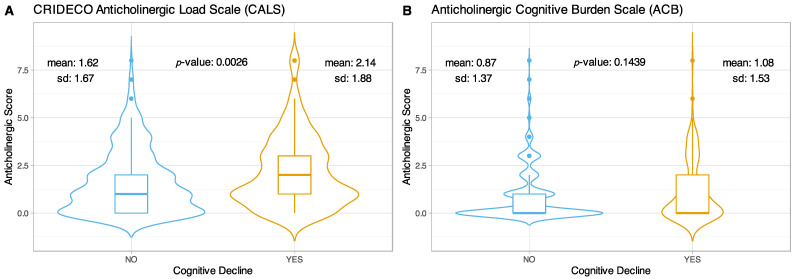
Violin plots comparing TAB scores in CI on the CALS (**A**) and ACB scales (**B**).

**Figure 4 jpm-12-00207-f004:**
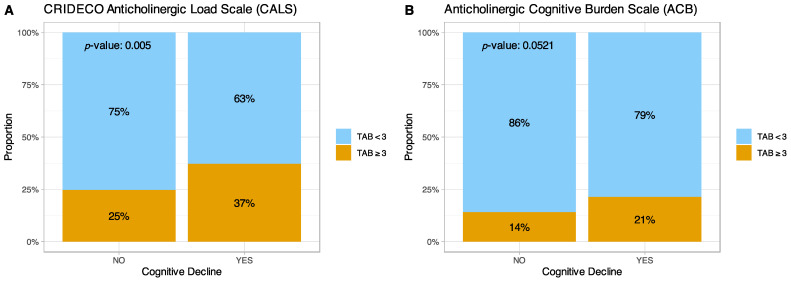
Graph comparing relevant anticholinergic burden and CI on the (**A**) CALS and (**B**) ACB scales. TAB = Total Anticholinergic Burden.

**Figure 5 jpm-12-00207-f005:**
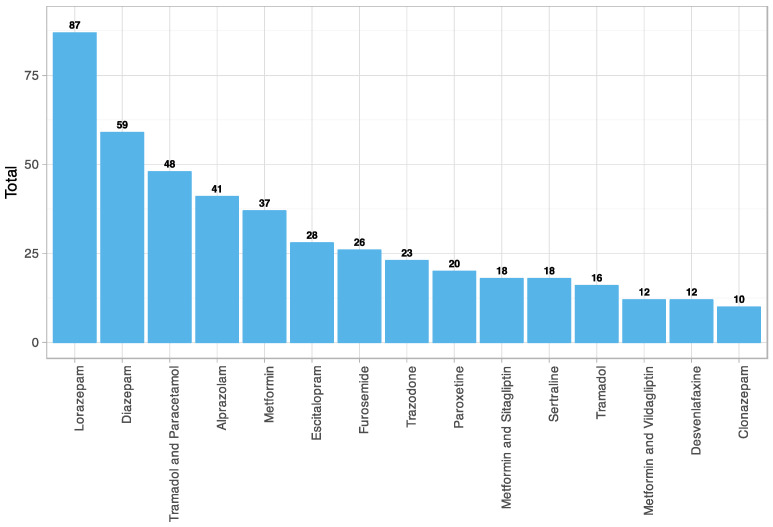
Anticholinergic drugs most used among patients with subjective memory complaints (CALS). N05BA06 = Lorazepam; N05BA01 = Diazepam; N02AJ13 = Tramadol and Paracetamol; N05BA12 = Alprazolam; A10BA02 = Metformin; N06AB10 = Escitalopram; C03CA01 = Furosemide; N06AX05 = Trazodone; N06AB05 = Paroxetine; A10BD07 = Metformin and Sitagliptin; N06AB06 = Sertraline; N02AX02 = Tramadol; A10BD08 = Metformin and Vildagliptin; N06AX23 = Desvenlafaxine; N03AE01 = Clonazepam.

**Table 1 jpm-12-00207-t001:** Low potency anticholinergics (Score 1).

Drug Name	ATC Code	Carnahan 2006	Rudolph 2008	Boustani 2008	Durán 2013	Salahudeen 2015	Kiesel 2018	Jun 2019
Aclidinium ^inh^	R03BB05						1	
Alimemazine *	R06AD01			1	2	1 or 2		1
Alprazolam	N05BA12	1		1	Disc	1 or 3	1	1
Alverine	A03AX08			1	Disc	1 or 2		0
Amisulpride	N05AL05							1
Ampicillin	J01CA01	1			Disc	1	1	0
Aripiprazole	N05AX12					1	1	1
Asenapine	N05AH05					1	1	
Atenolol	C07AB03	0		1		1	1	0
Azathioprine	L04AX01	1			Disc	1	1	0
Benazepril	C09AA07	0			Disc	1	1	0
Betaxolol	C07AB05	0			Disc	1	1	0
Bisacodyl	A06AB02	0			Disc	1	1	0
Bromocriptine	N04BC01	1			1 or 2	1	1	0
Bromperidol	N05AD06							1
Bupropion	N06AX12	0		1	Disc	1	1	1
Captopril	C09AA01	1		1	Disc	1	1	0
Cefamandole	J01DC03	1			Disc	1		0
Cefoxitin	J01DC01	1			Disc	1		0
Celecoxib	M01AH01	0			Disc	1	1	0
Cephalothin	J01DB03	1			Disc	1		0
Cetirizine	R06AE07	0	2		2	1 or 2	1	1
Cinnarizine	N07CA02							1
Chlordiazepoxide	N05BA02	1			1	1	1	1
Chlorthalidone	C03BA04	1		1	Disc	1	1	0
Citalopram	N06AB04	0			1	1	1	1
Clindamycin	J01FF01	1			Disc	1	1	0
Clonazepam	N03AE01	1			1	1	1	1
Clorazepate	N05BA05	1		1	Disc	1 or 3	1	1
Codeine	R05DA04	1		1	1 or 2	1 or 2	1	1
Colchicine	M04AC01	0			Disc	1 or 3		0
Cortisone	H02AB10	1			Disc	1		
Cycloserine	J04AB01	1			Disc	1		0
Cyclosporine	L04AD01	1			Disc	1		0
Desloratadine	R06AX27					1	1	1
Desvenlafaxine	N06AX23							1
Dexamethasone	H02AB02	1			Disc	1	1	0
Dextromethorphan	R05DA09	0			Disc	1	1	1
Diazepam	N05BA01	1		1	1	1	1	1
Digitoxin	C01AA04	1			1	1	1	
Digoxin	C01AA05	1		1	Disc	1 or 3	1	1
Diltiazem	C08DB01	1			Disc	1	1	0
Dipyridamole	B01AC07	1		1	Disc	1	1	0
Disopyramide	C01BA03	2		1	2	1 or 2		
Domperidone	A03FA03				1	1	1	0
Entacapone	N04BX02	0	1		1	1	1	0
Escitalopram	N06AB10	0			Disc	1	1	1
Estazolam	N05CD04	1			Disc	1		1
Famotidine	A02BA03	1			Disc	1	1	0
Fentanyl	N01AH01	1		1	1	1	1	1
Fexofenadine	R06AX26	0			2	2	1	0
Flunitrazepam	N05CD03				Disc		1	1
Flupentixol	N05AF01							1
Fluoxetine	N06AB03	1			1	1	1	1
Flurazepam	N05CD01	1			Disc	1	1	1
Fluvoxamine	N06AB08	1		1	1	1	1	1
Furosemide	C03CA01	1		1	Disc	1 or 3	1	1
Gentamicin	J01GB03	1			Disc	1	1	0
Glycopyrronium ^inh^	R03BB06						1	2
Guaifenesin	R05CA03	0			Disc	1	1	1
Haloperidol	N05AD01	0	1	1	1 or 2	1 or 2	2	1
Hydralazine	C02DB02	1		1	Disc	1	1	1
Hydrocodone	R05DA03	0			1 or 2	2		1
Hydrocortisone	H02AB09	1		1	Disc	1	1	1
Iloperidone	N05AX14					1		
Ipratropium ^inh^	R03BB01	0			3	3	1	
Isosorbide _mononitrate_	C01DA08	1		1	Disc	1	1	0
Isosorbide _dinitrate_	C01DA14	1		1	Disc	1	1	0
Ketorolac	M01AB15				1	1	1	0
Ketotifen	R06AX17	1			Disc	1		1
Levocetirizine	R06AE09					1	1	1
Levodopa—carbidopa	N04BA02	0	1		Disc	1	1	0
Lithium	N05AN01	0			1	1	1	0
Loperamide	A07DA03	1	2	1	1 or 2	1 or 2	2	1
Loratadine	R06AX13	0	2		1 or 2	1 or 2	1	1
Lorazepam	N05BA06	1			Disc	1	1	1
Lumiracoxib	M01AH06				Disc	1		
Mebeverine	A03AA04							1
Metformin	A10BA02	0			Disc	1	1	0
Methocarbamol	M03BA03		1		1	1	1	1
Methotrexate	L04AX03	0			Disc	1	1	0
Methylprednisolone	H02AB04	1			Disc	1	1	0
Metoclopramide	A03FA01	0	1		Disc	1	1	0
Metoprolol	C07AB02	0		1	0	1	1	0
Midazolam	N05CD08	1			Disc	1	1	1
Mirtazapine	N06AX11	0	1		1	1	1	1
Morphine	N02AA01	1		1	1	1	1	1
Naratriptan	N02CC02				Disc	1	1	0
Nefazodone	N06AX06	0			1	1		
Nifedipine	C08CA05	1		1	0	1	1	0
Nizatidine	A02BA04	1			Disc	1		
Oxazepam	N05BA04	1			Disc	1	1	
Oxycodone	N02AA05	1			1	1	1	1
Paliperidone	N05AX13					1	1	1
Pancuronium	M03AC01	1			Disc	1	1	
Phenelzine	N06AF03	1			1	1		
Phenobarbital	N03AA02	0			Disc	1	1	0
Piperacillin	J01CA12	1			Disc	1	1	0
Pramipexole	N04BC05	0	1		Disc	1	1	
Prednisolone	H02AB06	1			Disc	1	1	1
Prednisone	H02AB07	1		1		1	1	
Pridinol	M03BX03							1
Pseudoephedrine	R01BA02	0	2		Disc	2	1	0
Quinidine	C01BA01	0		1		1	1	
Risperidone	N05AX08	0	1	1	1	1	1	1
Rotigotine ^patch^	N04BC09						1	
Selegiline	N04BD01	0	1		Disc	1	1	0
Sertraline	N06AB06	1			0	1	1	0
Sumatriptan	N02CC01				Disc	1	1	0
Temazepam	N05CD07	1			1	1	1	1
Theophylline	R03DA04	1		1	1 or 2	1 or 2	2	1
Tiotixene	N05AF04	1	3		3	1 or 3		1
Tiotropium ^inh^	R03BB04						1	
Trandolapril	C09AA10	0			Disc	1	1	
Trazodone	N06AX05	0	1	1	1	1	1	1
Triamcinolone	H02AB08	1			Disc	1	1	0
Triamterene	C03DB02	1		1	Disc	1	1	0
Trimebutine	A03AA05							1
Triazolam	N05CD05	1			1	1	1	1
Valproic acid	N03AG01	1			Disc	1	1	0
Vancomycin	J01XA01	1			Disc	1	1	0
Venlafaxine	N06AX16	0			0	1	1	1
Warfarin	B01AA03	1			0	1	1	0
Ziprasidone	N05AE04		1		Disc	1	1	1
Zolmitriptan	N02CC03				Disc	1	1	0

* Alimemazine/Trimeprazine. Disc = discrepant for the authors. ^Inh^ = inhalative.

**Table 2 jpm-12-00207-t002:** Medium potency anticholinergics (Score 2).

Drug Name	ATC Code	Carnahan 2006	Rudolph 2008	Boustani 2008	Durán 2013	Salahudeen 2015	Kiesel 2018	Jun 2019
Amantadine	N04BB01	1	2	2	1 or 2	1 or 2	2	2
Baclofen	M03BX01	0	2		2	2	1	1
Carbamazepine	N03AF01	2		2	1 or 2	1 or 2	2	1
Cloperastine	R05DB21							2
Cimetidine	A02BA01	2	2	1	2	1 or 2	2	2
Cyclobenzaprine	M03BX08	2	2	2	1 or 2	1 or 2		2
Dosulepin	N06AA16				2	2		
Fluphenazine	N05AB02	1	3		3	1 or 3	1	
Loxapine	N05AH01	2		2	2	2	2	2
Maprotiline	N06AA21				Disc	3	2	
Meperidine *	N02AB02	2		2	2	2		2
Methadone	N07BC02				2	2	2	
Molindone	N05AE02	2		2	2	2		2
Nefopam	N02BG06					2		2
Olanzapine	N05AH03	1	2	3	1 or 2	1, 2 or 3	2	3
Oxcarbazepine	N03AF02	2		2	2	2	2	2
Paroxetine	N06AB05	1	1	3	1 or 2	1, 2 or 3	2	2
Perphenazine	N05AB03	1	3	3	Disc	1, 2 or 3	1	2
Pimozide	N05AG02	2		2	2	2	2	2
Prochlorperazine	N05AB04	1	2		1 or 2	1 or 2		
Promazine	N05AA03			3	2	2		
Propoxyphene	N02AC04	0			1 or 2	2		
Quetiapine	N05AH04	0	1	3	1 or 2	1, 2 or 3	2	2
Ranitidine	A02BA02	2	1		1 or 2	1 or 2	2	1
Tramadol	N02AX02	1			1 or 2	1 or 2	2	2
Triprolidine	R06AX07							2
Zotepine	N05AX11							2
Zuclopenthixol	N05AF05							2

* Meperidine/ Pethidine. Disc = discrepant for the authors.

**Table 3 jpm-12-00207-t003:** High potency anticholinergics (Score 3).

Drug Name	ATC Code	Carnahan 2006	Rudolph 2008	Boustani 2008	Durán 2013	Salahudeen 2015	Kiesel 2018	Jun 2019
Acepromazine	N05AA04				3	3		
Amitriptyline	N06AA09	3	3	3	3	3	3	3
Amoxapine	N06AA17			3	Disc	3		3
Atropine	A03BA01	3	3	3	3	3	3	3
Belladonna	A03BA04			2	3	2 or 3		3
Benzatropine *	N04AC01	3	3	3	3	3		3
Biperiden	N04AA02							3
Brompheniramine	R06AB01	3		3	3	3		3
Carbinoxamine	R06AA08	3		3	3	3		3
Carisoprodol	M03BA02	0	3		Disc	3		
Chlorphenamine *	R06AB04	3	3	3	3	3	3	3
Chlorpromazine	N05AA01	3	3	3	3	3		3
Chlorprothixene	N05AF03							3
Cimetropium _bromide_	A03BB05							3
Clemastine	R06AA04	3		3	3	3	3	3
Clomipramine	N06AA04	3		3	3	3	3	3
Clozapine	N05AH02	3	2	3	3	2 or 3	3	3
Cyproheptadine	R06AX02	2	3	2	3	2 or 3	3	2
Darifenacin	G04BD10	3		3	3	3	3	
Desipramine	N06AA01	3	2	3	3	2 or 3		
Dexbrompheniramine	R06AB06							3
Dexchlorpheniramine	R06AB02				3	3		3
Dicyclomine *	A03AA07	3	3	3	3	3		3
Difemerine	A03AA09							3
Diphenhydramine *	R06AA02	3	3	3	3	3	3	3
Doxepin	N06AA12	3		3	3	3	3	3
Doxylamine	R06AA09					3	1	3
Emepronium	G04BD01				3	3		
Fesoterodine	G04BD11					3	3	
Flavoxate	G04BD02	3		3	3	3	3	3
Homatropine	S01FA05				3	3		
Hydroxyzine	N05BB01	3	3	3	3	3	3	3
Hyoscyamine	A03BA03	3	3	3	3	3		3
Imipramine	N06AA02	3	3	3	3	3	3	3
Levomepromazine *	N05AA02	2		2	3	3	3	2
Meclozine *	R06AE05	3	3	3	3	3		3
Mequitazine	R06AD07							3
Nortriptyline	N06AA10	3	2	3	3	2 or 3	3	3
Opipramol	N06AA05				Disc	3	2	
Orphenadrine	N04AB02	3		3	3	3	3	3
Otilonium bromide	A03AB06							3
Oxybutynin	G04BD04	3	3	3	3	2 or 3	3	3
Pheniramine	R06AB05							3
Procyclidine	N04AA04	3		3	3	3	3	3
Promethazine	R06AD02	3	3	3	3	3	1	
Propantheline	A03AB05	3		3	3	2 or 3		
Propiverine	G04BD06					3	3	3
Protriptyline	N06AA11	3			3	3		
Pyrilamine *	R03DA12	3		3	3	3		3
Scopolamine *	A04AD01	3	3	3	3	3	3	3
Solifenacin	G04BD08					3	3	3
Thioridazine	N05AC02	3	3	3	3	3	3	3
Tiemonium iodide	A03AB17							3
Timepidium bromide	A03AB19							3
Tizanidine	M03BX02		3		3	3	3	2
Tolterodine	G04BD07	3	2	3	3	2 or 3	3	3
Trifluoperazine	N05AB06	1	3	3	Disc	1 or 3		
Trihexyphenidyl	N04AA01	3		3	3	3	3	3
Trimipramine	N06AA06	3		3	3	3	3	
Tropatepine	N04AA12				3	3		
Trospium	G04BD09					3	3	3
Valethamate	A03AX14							3

* Benzatropine/ Benztropine; * Dicyclomine/ Dicycloverine; * Diphenhydramine/ Dimenhydrinate; * Chlorphenamine/ Chlorpheniramine; * Levomepromazine/ Methotrimeprazine; * Meclozine/ Meclizine; * Pyrilamine/ Mepyramine; * Scopolamine/ Hyoscine. Disc = discrepant for the authors.

**Table 5 jpm-12-00207-t005:** Characteristics of patients in the CRIDECO study.

Characteristics of Study Participants (*n* = 512)	Cognitive Decline (*n* = 164)	No Cognitive Decline (*n* = 348)
Age, years (mean (SD))	74.67 (7.91)	68.09 (9.08)
Female (*n* (%))	123 (75%)	255 (73.27%)
BMI (mean (SD))	27.15 (3.85)	27.15 (3.90)
Drugs (mean (SD))	6.52 (3.31)	5.74 (3.28)
Mean TAB ACB score (mean (SD))	1.08 (1.53)	0.87 (1.37)
Mean TAB CALS score (mean (SD))	2.14 (1.88)	1.62 (1.67)
TAB CALS score ≥ 3 (*n* (%))	61 (37.20%)	86 (24.71%)
TAB ACB score ≥ 3 (*n* (%))	35 (21.34%)	49 (14.10%)
MIS test (mean (SD))	4.44 (2.31)	7.09 (0.99)
SPMSQ test (mean (SD))	2.99 (1.96)	0.73 (0.78)
SVF test (mean (SD))	10.05 (3.82)	16.10 (4.38)
Diabetes (*n* (%))	40 (24.39%)	77 (22.13%)
Hypertension (*n* (%))	100 (60.98%)	189 (54.31%)
Hypercholesterolemia (*n* (%))	82 (50%)	159 (45.69%)
Depression (*n* (%))	62 (37.80%)	100 (28.74%)

SD = Standard Deviation; BMI = Body Mass Index; TAB ACB = Total Anticholinergic Burden measured by Anticholinergic Cognitive Burden Scale; TAB CALS = Total Anticholinergic Burden measured by CRIDECO Anticholinergic Load Scale; MIS = Memory Impairment Screen; SPMSQ = Short Portable Mental Questionnaire; SVF = Semantic Verbal Fluency Test.

## Data Availability

The data used for this study is available upon request.

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
