# Peer review of "CRIDECO Anticholinergic Load Scale: An Updated Anticholinergic Burden Scale. Comparison with the ACB Scale in Spanish Individuals with Subjective Memory Complaints"

_jpm, 2022, doi:10.3390/jpm12020207_

Round 1

Reviewer 1 Report

Thank you for giving me the opportunity to review this manuscript. In this interesting study, authors designed a new anticholinergic scale and evaluated its association with cognitive impairment among individuals with subjective memory deficits;  compared with currently used measures of anticholinergic burden (e.g. ARS and ACB). The new scale, in contrast to previous ones, has shown to capture subjective cognitive impairment in patients with subjective cognitive deficits; this may really help identify individuals with future risk of cognitive deterioration who may need careful drug prescriptions and monitoring, as well as limiting prescription of anticholinergic medications. 

The topic of this manuscript is of interest given the weight of anticholinergic burden in geriatric populations and the risk associated with overprescribing and inappropriate prescribing of anticholinergic medications in frailer older individuals. The study is well conducted, methodology is appropriate, and both introduction and discussion are well presented and supported by previous studies. I have only a few minor concerns to be addressed before final publication:

1) Minor English corrections/typing errors

-Line 40: change the data with data

-Line 56: 'writing' is probably a typing error.

-Figure 1: change record excludes with record excluded' and 'elegilibility' with 'eligibility'. 

-Figure 3: improve figure legend.

-Figure 5: I think that using drug names instead of ATC codes for x axis ticks would render the figure more readable.

2)Conceptual suggestions:

-Line 100-102: the final part of the introduction may be decreased in length to avoid repetitions. You can directly say 'in order to explore the relationship

-Methodology: You mentioned artificial intelligence and machine learning algorithms. As it is a very wide chapter with heterogeneous techiques (supervised, unsupervised), I think you should extend the final part of statisical analysis with a brief description of the main techniques used to perform the analyses in this study.

-Table 5: Have you collected information about potential diseases related to cognitive impairment (cerebrovascular disease, cardiovascular disease, CKD, COPD, cancer, diabetes); can you add them to table 5? 

Author Response

Dear Reviewer:

We would like to thank you for your helpful improvements to the manuscript, we strongly appreciate your accurate review and kind words. The changes have been included in green color in the revised version and your comments have been discussed in the following lines.

1) Minor English corrections/typing errors

-Line 40: change the data with data

“The data” has been changed to “data”. Line 40. 

-Line 56: 'writing' is probably a typing error.

Thank you for your appreciation, the correct word is “writing confusion”, without the comma. It has been changed. Line 56.

-Figure 1: change record excludes with record excluded' and 'elegilibility' with 'eligibility'. 

Again, thanks for the appreciation, figure 1 has been modified correctly.

-Figure 3: improve figure legend.

We have removed the figure legend as it does not provide any additional information and maybe produce some confusion.

-Figure 5: I think that using drug names instead of ATC codes for x axis ticks would render the figure more readable.

In accordance with your suggestion, we have changed figure 5 to make it more readable.

2) Conceptual suggestions:

-Line 100-102: the final part of the introduction may be decreased in length to avoid repetitions. You can directly say 'in order to explore the relationship

We agree with your suggestion, it has been changed.

“in order to show the relationship between anticholinergic treatments and CI to explore the relationship between anticholinergic treatment and CI”. Lines 100-102.

-Methodology: You mentioned artificial intelligence and machine learning algorithms. As it is a very wide chapter with heterogeneous techniques (supervised, unsupervised), I think you should extend the final part of statistical analysis with a brief description of the main techniques used to perform the analyses in this study.

In accordance with your suggestion, for a better understanding, we have added some sentences in line 215 in the statistical treatment section referring the reader to the reference in our article [60] for a full explanation.

“A new artificial intelligence tool, developed by the research group, and based on an ensemble of different machine learning supervised techniques (such as bagging and boosting decision trees) was used to calculate the MCI probability result on performing the tests [60]” Lines 215 - 218.

-Table 5: Have you collected information about potential diseases related to cognitive impairment (cerebrovascular disease, cardiovascular disease, CKD, COPD, cancer, diabetes); can you add them to table 5? 

In accordance with your suggestion, we have added diabetes, hypertension, hypercholesterolemia, and depression to the table 5.  For the rest of the variables such as cancer, COPD, CKD we do not have information in our database.

Finally to mention, that we have also added to certify the English language has been reviewed by an external English proofreading service. Line 494.

Thanks and best regards

Reviewer 2 Report

The authors have addressed a very important issue of anticholinergic burden in elderly patients using a novel approach.

The study is well conceptualized and written in standard English language.

The methodology is sound and results well illustrated.

The discussion is rich and cited relevant literature in the field.

Author Response

Dear reviewer:

We would like to thank you for your comments on the manuscript, we greatly appreciate your review.

We have made a revision to the English, as suggested by you and reviewer 1. And, we have also added to certify the English language that the article has been reviewed by an external English proofreading service. Line 494.

Thanks and best regards